

# Gobal and regional chemical influence of sprites: Reconciling modeling results and measurements

Francisco J. Pérez-Invernón[1], Francisco J. Gordillo-Vázquez[1], Alejandro Malagón-Romero[2], and Patrick Jöckel[3]

[1]Instituto de Astrofísica de Andalucía (IAA), CSIC, PO Box 3004, 18080 Granada, Spain
[2]Centrum Wiskunde & Informatica (CWI), Amsterdam, The Netherlands
[3]Deutsches Zentrum für Luft- und Raumfahrt, Institut für Physik der Atmosphäre, Oberpfaffenhofen, Germany

**Correspondence:** Francisco J. Pérez-Invernón (fjpi89@gmail.com)

**Abstract.** Mesospheric electrical discharges, known as sprites, formed by fast-propagating streamers, have been shown to create localized enhancements of atmospheric constituents such as N, O, $NO_x$, $N_2O$, and $HO_x$, as indicated by both, modeling results and space-based measurements. In this study, we incorporate the occurrence rate of sprites into a chemistry-climate model using meteorological parameters as a proxy. Additionally, we introduce the injection of chemical species by sprites into

the model, based on electrodynamical modeling of individual sprite streamers and observations from space.

Our modeling results show a good agreement between the simulated sprite distribution and observed data on a global scale. While the global influence of sprites on the atmospheric chemistry is found to be negligible, our findings reveal their measurable chemical influence at regional scale, particularly for the concentration of $HNO_3$ and $HNO_4$ within the mesosphere. The simulations also suggest that sprites could be responsible for the observed $NO_2$ anomalies at an altitude of 52 km above

thunderstorms, as reported by MIPAS. Finally, a projected simulation reveals that the occurrence rate of sprites could increase at a rate of 14% per 1 K rise in the global temperature.

## 1   Introduction

In 1925, Wilson (1925) predicted the existence of electrical discharges (nowadays called sprites) above thunderstorms, which were later confirmed by Franz et al. (1990). Sprites are a category of Transient Luminous Events (TLEs) that take place

at altitudes ranging from 40 km to 90 km. They are initiated by the ionization resulting from the quasi-electrostatic field component of lightning discharges, as described by studies such as, for instance, Pasko et al. (1997); Stenbaek-Nielsen et al. (2000); Pasko et al. (2012). The quasi-electrostatic field primarily generates electromagnetic radiation within the Extremely Low Frequency (ELF) and the Very Low Frequency (VLF) electromagnetic spectra, which result from continuing currents during discharges lasting several tens or hundreds of milliseconds (Brook et al., 1962). Consequently, sprites are commonly

observed simultaneously with a discernible ELF and/or VLF signal, emitted by lightning flashes characterized by continuing currents (Greenberg et al., 2009; Inan et al., 2010; Lu et al., 2017).

Sprites consist of fast-propagating streamers followed by long-standing luminous structures known as beads and glows. These events typically last between 1 and 100 milliseconds (Liu, 2010; Luque and Gordillo-Vázquez, 2010, 2011; Luque et al.,



2016; Malagón-Romero et al., 2020). The primary sources of optical emissions in sprite streamers, beads, and glows originate
from various molecular components, including the first and second positive systems of molecular neutral nitrogen, the first
negative system of molecular nitrogen ions, the Meinel band of molecular nitrogen ions, and the Lyman-Birge-Hopfield (LBH)
band of molecular neutral nitrogen (Armstrong et al., 1998; Chen et al., 2003; Kuo et al., 2005; Stenbaek-Nielsen et al., 2007;
Kanmae et al., 2007; Šimek, 2014; Sato et al., 2015; Hoder et al., 2016; Ihaddadene and Celestin, 2017; Gordillo-Vázquez
et al., 2018; Pérez-Invernón et al., 2018b).

Observing sprites is challenging due to their short duration. However, ground-, space- and aircraft-based instruments have
been successful in detecting sprites, providing valuable information about their occurrence (Armstrong et al., 1998; Stenbaek-
Nielsen et al., 2007; Gordillo-Vázquez et al., 2018; Arnone et al., 2020). To overcome the limitations of direct observations,
some researchers have proposed using ELF and VLF lightning measurements from flashes with continuing currents as a
proxy indicator for the occurrence of sprites (Füllekrug and Constable, 2000; Sato and Fukunishi, 2003; Andrey et al., 2022).
Following this approach, Ignaccolo et al. (2006) estimated a global occurrence rate of 2.8 sprites per minute with an accuracy
factor of ∼2-3. Chen et al. (2008) used satellite-based optical observations from the Imager of Sprites and Upper Atmospheric
Lightning (ISUAL) experiment aboard the FORMOSAT-2 satellite to report a global occurrence rate of 0.5 sprites per minute,
while also providing information about the polarity of the lightning parents and the distribution of sprites over land and ocean
(Zhang et al., 2022). More recently, Andrey et al. (2022) estimated a global occurrence rate of about 0.6 sprites per minute
based on global measurements of the energy radiated by cloud-to-ground (CG) lightning reported by the World Wide Lightning
Location Network (WWLLN).

   Electrodynamical and chemical models of sprites suggest a significant local production of $NO_x$ (NO+$NO_2$), $N_2O$ and $HO_x$
(H+OH+$HO_2$) in the mesosphere (from about 40 km upwards) and the lower-ionosphere (Sentman et al., 2008; Gordillo-
Vázquez, 2008, 2010; Gordillo-Vázquez et al., 2012; Evtushenko et al., 2013; Winkler and Nothold, 2014; Parra-Rojas et al.,
2015; Pérez-Invernón et al., 2020; Winkler et al., 2021). According to Sentman et al. (2008), they calculated a production of
$5 \times 10^{19}$ molecules of NO per single streamer in sprites between altitudes of 65 km and 75 km by using a chemical model of
sprites. Enell et al. (2008) reported a production of $3 \times 10^{22}$ to $3 \times 10^{23}$ NO molecules per complete sprite. Pérez-Invernón
et al. (2020) calculated a production of $N_2O$ and NO molecules per sprite between 68 and 75 km altitude of $2 \times 10^{19}$ and $10^{21}$
molecules, respectively. Winkler et al. (2021) modeled the production of $HO_2$ by sprite-streamers and found that they could
produce about $10^{20}$ molecules of $HO_2$ between 70 and 80 km altitudes. Finally, Malagón-Romero et al. (2023) extended the
electrodynamical model of sprite-streamers of Pérez-Invernón et al. (2020) to estimate a production of $7.5 \times 10^{18}$ molecules
NO, $2.6 \times 10^{18}$ molecules of $NO_2$, $2.6 \times 10^{18}$ molecules of $N_2O$ and a removal of $3.1 \times 10^{22}$ molecules of $O_3$ by sprite-
streamers between 49.75 km and 50 km.

   The possibility of sprites producing $NO_x$ and $HO_x$ in the mesosphere has motivated several attempts of measuring the chemi-
cal production of sprites to determine their chemical role in the atmosphere. Arnone et al. (2008) combined $NO_2$ measurements
obtained from the Michelson Interferometer for Passive Atmospheric Sounding (MIPAS) with lightning data sourced from the
World Wide Lightning Location Network (WWLLN) for the period August to December 2003. They conducted a search for
anomalies in nighttime measurements of $NO_2$ mixing ratios (at about 22.00 local time) at altitudes of 47 km, 52 km, and 60 km



above thunderstorms. This search was limited to latitudes within the range of 30°S to 20°N and over a instantaneous field of
view with a footprint of 30 km in longitude × 1200 km in latitude at 52 km. To examine the relationship between lightning
activity and NO$_2$ mixing ratios, they generated five sets of NO$_2$ measurements based on the accumulation of lightning events
prior to the MIPAS overpass. The first set consisted of measurements taken in the absence of lightning flashes up to 60 minutes
before the MIPAS overpass, while the remaining sets were similar, but included data from 10, 20, and 30 minutes prior to the
overpass. They reported a maximum positive anomaly of the NO$_2$ mixing ratio of + 1 ppbV 20 minutes after the occurrence of
lightning at 52 km. Subsequently, Arnone and Dinelli (2016) extended their investigation up to April 2004, corroborating the
presence of an elevated mixing ratio of NO$_2$ above thunderstorms. However, when the analysis was further expanded to encom-
pass the entire MIPAS2D dataset (Dinelli et al., 2010), no significant augmentation in the NO$_2$ mixing ratio was discernible at
an altitude of 52 km above thunderstorms. These results collectively suggest that the chemical disturbance induced by sprites
in the mesosphere resides on the cusp of current detection capabilities. Following these measurements, Arnone et al. (2014) in-
troduced a parameterization scheme into the Whole Atmosphere Community Climate Model (WACCM) to explore how sprites
influence the chemistry of the mesosphere. They incorporated the injection of sprite-generated NO$_x$ based on the latest findings
from sprite-streamer modeling, simulating a global rate of 2-3 sprites per minute. Their study encompassed both, summer and
winter conditions, involving simulations covering 40 days each. Their results revealed an elevation of 0.015-0.15 ppbv of the
NO$_x$ mixing ratio at 70 km altitude in tropical regions, although this effect became insignificant on a global scale. Furthermore,
they identified a potential localized increase of up to tens of percent of the NO$_x$ mixing ratio within the altitude range of 60 to
km. This increase, while potentially detectable by current instruments like MIPAS, remains a localized phenomenon.

Sentman et al. (2003) reported that sprites produce no distinctive OH emissions at the 2% background brightness level,
indicating an upper estimate in the perturbation of OH by sprites. Recently, Yamada et al. (2020) documented a notable
increase of the HO$_2$ mixing ratio in three regions following the incidence of sprites. They used limb spectral measurements
reported by the Submillimeter-Wave Limb-Emission Sounder (SMILES) and estimated that a single sprite could produce up to
$10^{25}$ molecules of HO$_2$ between 75 km and 80 km altitude, which is considerably larger than the production of HO$_2$ estimated
by Winkler et al. (2021). An injection of $10^{25}$ molecules of HO$_2$ per sprite implies that sprites could represent up to the 1%
of the global source of nighttime background HO$_2$ in the upper mesosphere. Nevertheless, there remains uncertainty regarding
whether measurements of sprite chemical activity (Yamada et al., 2020) might be influenced, either partially or entirely, by the
chemical production of lightning-induced electron precipitation in the mesosphere (Xu et al., 2021).

In this study, we present the first parameterization of sprites based on proxy meteorological parameters for sprite activity.
We implement this parameterization in the Modular Earth Submodel System (MESSy) for usage within the European Center
HAMburg general circulation model (ECHAM) / MESSy Atmospheric Chemistry (EMAC) (Jöckel et al., 2010, 2016). The
parameterization of sprites is based on the parameterization of the occurrence rate of Long-Continuing-Current (LCC) lightning
developed by Pérez-Invernón et al. (2022), enabling us to investigate the global seasonal variability in the occurrence of sprites
as well as their sensitivity to climate change. In addition, we introduce in the parameterization the injection of NO$_x$, N$_2$O and
HO$_2$ by sprites as well as the direct depletion of O$_3$ between 45 and 80 km altitude by using the modeling results of Pérez-
Invernón et al. (2020); Winkler et al. (2021) and Malagón-Romero et al. (2023). In turn, we compare the simulated NO$_2$ mixing





ratio resulting from model simulations of sprites with the nighttime positive anomalies in the $NO_2$ mixing ratio reported by MIPAS above thunderstorms (Arnone et al., 2014; Arnone and Dinelli, 2016) to assess the potential influence of sprites on these measurements.

## 2 Model

### 2.1 The EMAC model

The numerical chemistry-climate model EMAC couples ECHAM5 (Roeckner et al., 2006) with the MESSy framework to
connect various multi-institutional computer codes, referred to as MESSy submodels (Jöckel et al., 2010, 2016). The submodels are employed to depict processes within the troposphere and middle atmosphere, as well as their interactions with oceans, land, and external factors originating from anthropogenic emissions.

The model is operated at T42L90MA resolution, i.e. with a triangular trunctaion of the spectral resolution at wave number 42, corresponding to a quadratic Gaussian grid with a resolution of 2.8° in both latitude and longitude. It comprises 90 vertical
levels, extending up to the 0.01 hPa pressure level, and employs a time step length of 720 seconds, as described by Jöckel et al. (2016). Additionally, the Tiedtke/Nordeng convection scheme (Tiedtke, 1989; Nordeng, 1994) implemented within the CONVECT submodel is utilized.

The LNOX submodel of MESSy is used to calculate the occurrence rate of sprite-triggering LCC lightning flashes. The LNOX submodel calculates the total lightning flash frequency, the LCC lightning flash frequency, and the production of $NO_x$
by lightning from several lightning parameterizations selected by the user (Tost et al., 2007) and by fixing a scaling factor that results in a lightning occurrence rate of ∼45 flashes per second globally (Christian et al., 2003; Cecil et al., 2014). For the present study, we used the parameterization of lightning flashes producing the best comparison between the simulated and the observed LCC lightning flash density (Bitzer, 2017), i. e., the lightning parameterization based on the Cloud Top Height (CTH) by Price and Rind (1992) for land, combined with a parameterization of lightning that used the updraft strength at 440 hPa
pressure level (Allen and Pickering, 2002) for ocean, with a global scaling factor of 1.13 (Pérez-Invernón et al., 2022). The submodel LNOX calculates the occurrence rate of LCC flashes with a continuing current longer than 18 ms (LCC(>18 ms) lightning) from the updraught mass flux by employing the parameterization for the ratio of LCC to total lightning, as developed by Pérez-Invernón et al. (2022).

### 2.2 Parameterization of sprites

A new submodel named SPRITES is developed to include the parameterization of sprites in MESSy v2.54 and will be implemented in the submodel LNOX in future versions of MESSy. The submodel SPRITES calculates the sprite flash density and the production of NO, $NO_2$, $N_2O$ and $HO_2$ and, in turn, the depletion of $O_3$ by sprites between 45 km and 75 km altitude.



### 2.2.1 Sprite occurrence

The occurrence rate of sprites in the submodel SPRITES is implemented by using the calculation of lightning density from the
LNOX submodel. Lightning flashes with continuing currents, such as LCC(>18 ms) lightning, can produce a significant quasi-electrostatic field in the mesosphere (Gamerota et al., 2011) and, in turn, trigger the inception of sprites [e.g., (Pasko et al., 1997; Stenbaek-Nielsen et al., 2000; Pasko et al., 2012)]. Therefore, the LCC(>18 ms) lightning density computed by the LNOX submodel of MESSy (Tost et al., 2007; Pérez-Invernón et al., 2022) is used as a proxy for the occurrence of sprites. In addition, we imposed that nighttime sprites can only be produced after sunset, as the absence of solar radiation during nighttime
contributes to reduce the ionization of the lower ionosphere and, in turn, favors the electric breakdown of the air that triggers the inception of sprites (Pérez-Invernón et al., 2016). Finally, we imposed that only 20% of the nighttime LCC(>18 ms) lightning flashes have the potential to trigger sprites, following the ELF measurements of lightning with continuing current reported by Füllekrug and Constable (2000). The assumed 20% is an upper limit, as Füllekrug and Constable (2000) reported that between 5 and 20% of the measured lightning flashes had the potential to produce air electric breakdown at sprite altitude.
However, our approach lacks consideration for sprites triggered by lightning without continuing currents, which may lead to an underestimation of sprite occurrences (Inan et al., 2010). Finally, it is worth noting that Greenberg et al. (2009) found that approximately 33% of the 15 sprites analyzed in Europe were produced by lightning strikes unaccompanied by associated ELF transients. Similarly, Lu et al. (2017) reported that 15% of the 247 recorded sprites in North America were the result of negative cloud-to-ground flashes without detectable continuing currents.

### 2.2.2 Chemical influence of sprites

The submodel SPRITES introduces the chemical influence of sprites in the mesosphere by multiplying the calculated sprite density and the production/destruction of chemical species by single sprites. The $HO_2$ molecules produced by sprites are homogeneously distributed between 70 km and 75 km altitude. The submodel's namelist allows the user to choose a total injection of $10^{20}$ or $10^{25}$ molecules of $HO_2$ per sprite, based on electrodynamic modeling results (Winkler et al., 2021) and
measurements (Yamada et al., 2020), respectively.

The injection of NO, $NO_2$ and $N_2O$ molecules, as well as the direct depletion of $O_3$ molecules implemented in the SPRITE submodel are based on modeling results of single sprite streamers in the mesosphere-lower thermosphere (67-75 km) and in the lower mesosphere (49.75-50 km) by Pérez-Invernón et al. (2020) and Malagón-Romero et al. (2023), respectively. The production of chemical species by sprites between the ranges of altitudes from 45 km to 49.75 km and from to 50 km and 67 km
is estimated by following the approach developed by Malagón-Romero et al. (2023), i. e., by interpolating and extrapolating from the results by Pérez-Invernón et al. (2020) and Malagón-Romero et al. (2023). Following this approach, Malagón-Romero et al. (2023) obtained $6.2 \times 10^{20}$ NO molecules, $2.6 \times 10^{19}$ $NO_2$ molecules and $1.7 \times 10^{20}$ $N_2O$ molecules injected by a single sprite streamer in the range of altitudes between 45 km and 75 km, while they reported a removal of $3.1 \times 10^{23}$ $O_3$ molecules. In addition, we apply a conversion factor between the chemical injection by a single sprite and by a complete sprite in EMAC.
Pérez-Invernón et al. (2020) reported a scaling factor ranging between 18 and 50. We have updated the estimation of this



**Table 1.** Overview of the performed simulations.

| Simulation | Mode | Years | Sprites | Sprites-chemistry |
|---|---|---|---|---|
| SPRI | Dynamical. Nudged towards ERA-Interim reanalysis | 2000-2009 | Yes | No |
| RCP6.0 | Active chemistry. Projection RCP6.0 | 2090-2095 | Yes | No |
| BASE | Active chemistry. Nudged towards ERA-Interim reanalysis | 2000-2001 | No | No |
| CTRL | Active chemistry. QCTM from BASE | 2000-2001 | No | No |
| SPRI-M | Active chemistry. QCTM from BASE | 2000-2001 | Yes | Yes ($HO_x$ by Malagón-Romero et al. (2023)) |
| SPRI-SMI | Active chemistry. QCTM from BASE | 2000-2001 | Yes | Yes ($HO_x$ by Yamada et al. (2020)) |

scaling factor by using recent detections of sprites by the Atmosphere-Space Interactions Monitor (ASIM). Gomez Kuri (2021) reported the detection of a sprite on July 10, 2019 by combining optical measurements of ASIM and ELF measurements from ground-based sensors. We have integrated the optical signal detected by ASIM in the range of wavelengths between 180 nm and 230 nm (ASIM photometer 2) during the 0.85 ms after the onset of the first and the second peaks associated with the

sprite event at times 21:53:17.554 and 21:53:17.563 (Gomez Kuri, 2021, Figure 4.16), obtaining observed photometric fluxes of $10^{-10}$ J m$^{-2}$ and $1.1 \times 10^{-10}$ J m$^{-2}$, respectively. The synthetic flux of the streamer simulated by Pérez-Invernón et al. (2020) that would be observed by ASIM during 0.85 ms in the range of wavelengths between 180 nm and 230 nm is $10^{-12}$ J m$^{-2}$. Therefore, we can assume that a complete sprite emits about 100 times more photons than the simulated single sprite streamer. Therefore, we used a factor of 100 to convert the simulated injection of molecules by a single sprite streamer into the

injection of molecules by a complete sprite.

### 2.3 Simulation set-up

Table 1 shows the overview of the performed simulations. Firstly, a purely dynamical simulation (SPRI) covering the present day climatic state is performed during the period 2000-2009 by nudging the model towards ERA-Interim reanalysis meteorological fields (Dee et al., 2011) to evaluate the sprite frequency parameterization. A projection simulation (RCP6.0) covering

the years 2090-2095 is performed under Representative Concentration Pathway 6.0 (RCP6.0) in order to evaluate the sensitivity of sprites under climate change. We consider the years 2000 and 2090 as the spin-up phases. The RCP6.0 simulation is established following the simulation RC2-base-04 of Jöckel et al. (2016); Pérez-Invernón et al. (2023). The sea surface temperatures (SSTs) and the sea-ice concentrations (SICs) are prescribed from simulations with the Hadley Centre Global Environment Model version 2 - Earth System (HadGEM2-ES) Model (Collins et al., 2011; Bellouin et al., 2011). Projected greenhouse

gases and $SF_6$ mixing ratios are included from Eyring et al. (2013). Anthropogenic emissions are taken from monthly values provided by Fujino et al. (2006) for the RCP6.0 scenario. The chemical influence of sprites in the atmosphere has been deactivated in this simulation in order to avoid unexpected perturbations in the chemistry. We refer to Jöckel et al. (2016) for more details about the simulations set-up. Pérez-Invernón et al. (2023) obtained a temperature increase of 4 K between present day and 2091-2095 by using the same setup.

In turn, a set of simulations with active chemistry are performed to evaluate the chemical role of sprites in the atmosphere in the Quasi Chemistry-Transport Model (QCTM) mode proposed by Deckert et al. (2011) to ensure that small chemical perturba-





tions do not alter the simulated meteorology by introducing noise. Firstly, a two years simulation nudged towards ERA-Interim reanalysis meteorological fields (Dee et al., 2011; ECMWF, 2011) and without sprites is performed (BASE). The simulation is set-up following the simulation with interactive chemistry RC1SD-base-07 of Jöckel et al. (2016). The sea surface temperatures (SSTs) and the sea-ice concentrations (SICs) are used from ERA-Interim reanalysis data (Dee et al., 2011). The chemical kinetics is simulated by using the submodel MECCA (Module Efficiently Calculating the Chemistry of the Atmosphere) by Sander et al. (2019). Next, we conduct a present-time simulation in the QCTM mode, featuring active chemistry but excluding sprites (CTRL). This simulation is generated utilizing inputs for radiation calculations and methane oxidation from the BASE simulation. This approach allows us to separate dynamics from chemistry while maintaining consistent meteorological conditions, effectively suppressing meteorological variability. Lastly, we perform two additional present-day simulations, both in the QCTM mode and incorporating sprites. These simulations are denoted as SPRI-M and SPRI-SMI, respectively. In the SPRI-M simulation we have used modeling results of single sprite streamers (Pérez-Invernón et al., 2020; Malagón-Romero et al., 2023) to set the injection chemical species, while in the simulation SPRI-SMI we have modified the injection of $HO_x$ accordingly to measurements from SMILES (Yamada et al., 2020) (see Section 2.2.2). Comparison of simulations CTRL, SPRI-M and SPRI-SMI allow us to determine the chemical role of sprites in the mesosphere. In this set of simulations, we consider the year 2000 as the spin-up phase.

## 3 Results and discussion

### 3.1 Geographical distribution of sprites

Details on the simulated global frequency of sprites and lightning are summarized in Table 2, while the simulated annually averaged sprite flash densities for present day and 2091-2095 are shown in Figure 1. We obtained a global sprite occurrence rate of 1.66 sprites per minute in 2000-2009, which is above the value reported by Chen et al. (2008) (0.5 sprites per minute) and below the value reported by Ignaccolo et al. (2006), ranging between 2 and 3 sprites per minute. In turn, we obtained a global occurrence rate of 2.55 sprites per minute in 2091-2095, which represents an increase of 54% (14% increase per 1 K increase in the global temperature between the period 2091-2095 and 2000-2009). The simulated increase of the global occurrence rate of sprites between present day and 2091-2095 is approximately similar to the simulated increase of the global occurrence rate of total lightning (47%).

The simulated latitudinal distribution of sprites has a pronounced peak between 10°N and 20°N and another less pronounced peak between 30°N and 40°N (see Figure 2(a)), in agreement with the climatology reported by (Zhang et al., 2022, Figure 5) and (Lu et al., 2022, Figure 2(b)) from ISUAL measurements. However, the simulated latitudinal distribution of sprites presents a main peak between 10°S and 20°S that was not detected by ISUAL. The disagreement between the observed and the simulated latitudinal distribution of sprites is influenced by the overestimation of simulated sprites over land between 10°S and 20°S (Figure 2(b)), i.e. in South America and Southeastern Asia. In turn, we found a good agreement between the simulated and the observed (Zhang et al., 2022; Lu et al., 2022) latitudinal distribution of sprites over ocean (Figure 2(c)), except for the observed peak near 40°N that is less pronounced in the simulations. We found a sprite land-ocean contrast of 0.9 : 1.1 and 1 : 1



**Table 2.** Occurrence rate of lightning and sprites from the simulations SPRI and RCP6.0.

|                                            | 2001-2009          | 2091-2095          |
| ------------------------------------------ | ------------------ | ------------------ |
| Global occurrence rate of lightning flashes | 45 flashes s$^{-1}$ | 66 flashes s$^{-1}$ |
| Global occurrence rate of sprites          | 1.66 sprites min$^{-1}$ | 2.55 sprites min$^{-1}$ |
| Lightning land-ocean contrast              | 1 : 1              | 1 : 1              |
| Sprites land-ocean contrast                | 0.9 : 1.1          | 1 : 1              |

for present day and 2091-2095, respectively. Chen et al. (2008) reported a sprite land-coast-ocean contrast of 4.7 : 3.2 : 1. The horizontal resolution of our simulations (2.8° × 2.8° quadratic Gaussian grid in latitude and longitude) does not allow us to distinguish between coast and land and coast and ocean. The parameterization of Andrey et al. (2022) produced 41.4% sprites over land and 58.6% over ocean, while we obtained 45% and 55%, respectively.

The simulated global density of sprites over land in present day simulations is in fairly good in agreement with the obser-
vations reported by Chen et al. (2008); Zhang et al. (2022), showing hotspots in Middle Africa, South America, Eastern North America, the Tornado Alley of North America, Western Europe and Southeastern Asia. The simulation produced an overestimation of the sprite density in Brazil, Southern Africa and China that can be explained by the low accumulative observation time of ISUAL in these regions (Chen et al., 2008). The high occurrence of sprites in the Mediterranean Sea and Western Europe is in agreement with the European climatology of sprites reported by Yair et al. (2015) and Arnone et al. (2020).

Figure 3 shows the averaged sprite flash density during the seasons December-January-February (DJF), March-April-May (MAM), June-July-August (JJA) and September-October-November (SON) during 2000-2009. The maximum occurrence of sprites is reached during summer (DJF in the Southern Hemisphere, JJA in the Northern Hemisphere). During MAM the global density of sprites is shifted towards the Equatorial region and the Southern Hemisphere, while it is equally distributed between both hemispheres in SON. Sato and Fukunishi (2003) reported the global seasonal distribution of sprites based on ELF
measurements for DJF, JJA and SON. There is a good agreement between the simulated and the reported sprite distribution in JJA near the Equator and in Southeastern Asia, while the present parameterization underestimates the production of sprites in North and South America. In DJF, the simulation is in agreement with the climatology of sprites based on ELF measurements (Sato and Fukunishi, 2003) in all regions except for North America. Finally, we have found a good agreement between the simulations and the ELF based climatology during SON.

**3.2 Global chemical influence of sprite-streamers in the mesosphere-lower thermosphere**

Figure 4 shows the simulated influence of sprites in the annually and globally averaged vertical profiles of NO$_x$=NO+NO$_2$, N$_2$O, HO$_x$=H+OH+HO$_2$, O$_3$, HNO$_3$ and HNO$_4$ by assuming that single sprites inject $10^{20}$ HO$_2$ molecules (Winkler et al., 2021). The obtained small variations between simulations with and without sprites clearly show that the influence of sprites is negligible on the global scale. The maximum effect of sprites in the vertical profiles of NO$_x$ and N$_2$O are located in the
upper mesosphere, where the background abundance of these species is low. The found contribution of sprites to the global





**Figure 1.** Simulated annually averaged sprite flash density in sprites per squared kilometer and day during 2001-2009 from the SPRI simulation (a) and during 2091-2095 from the RCP6.0 simulation (b). We annotate in boxes the annually averaged occurrence rate of sprites per minute.




**Figure 2.** The solid lines depict the simulated annual average latitudinal sprite flash density, measured in sprites per square kilometer per day, spanning the years 2001 to 2009 (SPRI simulation), encompassing both, land and ocean regions. The dashed lines represent the total number of pure sprites and sprites accompanied by halos observed by ISUAL, as illustrated by (Lu et al., 2022, Figure 2(b)). The observations have been interpolated to the corresponding latitudes of the simulation results.



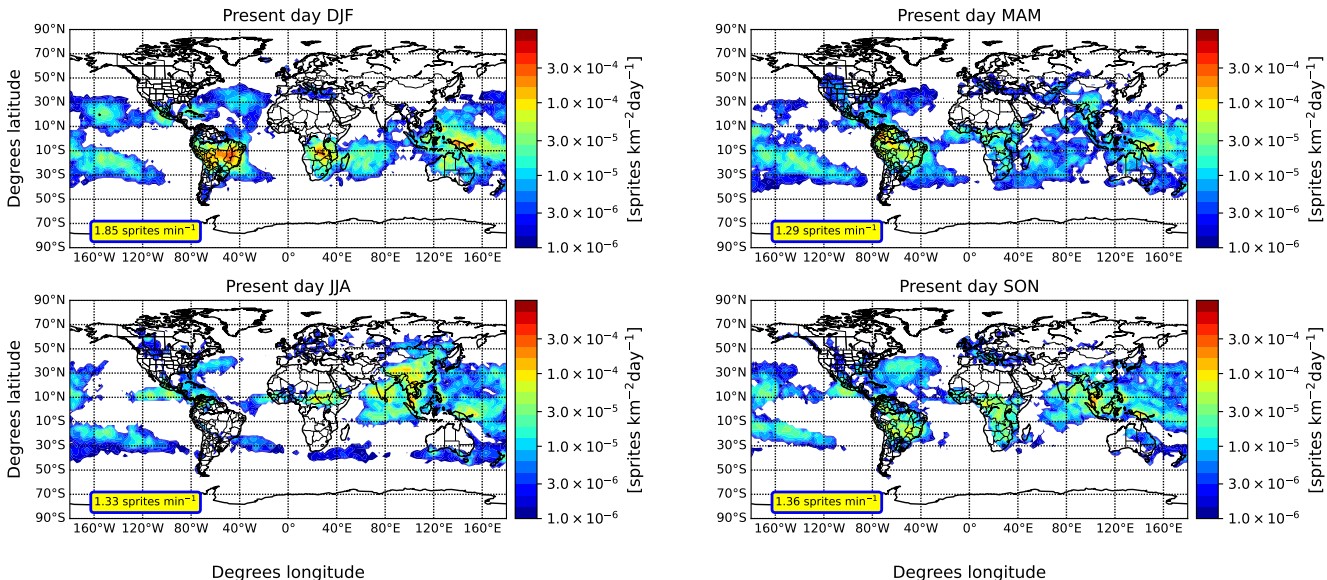

**Figure 3.** Simulated seasonally averaged sprite flash density in sprites per squared kilometer and day during 2001-2009 from the SPRI simulation. We annotate in boxes the seasonally averaged occurrence rate of sprites per minute.

amount of $N_2O$ in the upper mesosphere is about 0.002%, in agreement with previous estimates by Pérez-Invernón et al. (2020) of 0.003% from electrodynamical simulations of streamers. The background $HO_x$ mixing ratio in the upper mesosphere is slightly enhanced in the presence of sprites and it is dominated by the conversion of $HO_2$ into OH and H. The amount of $HO_x$ is reduced in the middle and lower mesosphere as a consequence of the conversion of $HO_2$ and OH into the nitrogen reactive

compounds $HNO_3$ and $HNO_4$ produced by the injection of $NO_x$. As a consequence, the mixing ratio of $HNO_3$ and $HNO_4$ increased in the upper mesosphere. In turn, the injection of NO and $HO_2$ by sprites produces an enhancement in the upper-mesospheric background $O_3$ mixing ratio, while the net contribution of sprites to $O_3$ in the middle and the lower mesosphere is negative.

We show in Figure 5 the influence of sprites on the annually and globally averaged vertical profiles of $NO_x=NO+NO_2$, $N_2O$,

$HO_x=H+OH+HO_2$ and $O_3$ by assuming that single sprites inject $10^{25}$ $HO_2$ molecules (Yamada et al., 2020). There are some relevant differences between the enhancements of $NO_x$ when introducing $10^{25}$ $HO_2$ molecules instead of $10^{20}$ $HO_2$ molecules per sprite. The injected $HO_2$ produces a modification of the contribution of NO and $NO_2$ to the total $NO_x$. The $HO_2$ reacts with NO, producing a decrease in the concentration of NO in the upper mesosphere and an increase of about 0.06% in $NO_2$. The injection of $10^{25}$ $HO_2$ molecules per sprite produces a large $HO_x$ mixing ratio between 60 km and 80 km altitude, still too

low to be considered a significant source of $HO_x$ at a global scale. However, the conversion of OH and $HO_2$ into the reactive nitrogen compounds $HNO_3$ and $HNO_4$ in combination with $NO_x$ led to a 0.3% and a 0.04% enhancement in the mixing ratio of $HNO_4$ and $HNO_3$ in the upper mesosphere, respectively. In turn, the influence of sprites on $O_3$ is different, when introducing



$10^{25}$ HO$_2$ molecules per sprite. The injected HO$_2$ contributes to a decrease of the background mixing ratio of O$_3$ in the upper mesosphere (about -0.002% at a global scale). The depletion of O$_3$ by HO$_x$ can be due to the enhancement of NO$_2$, which
contributes to deplete the mixing ratio of O$_3$.

We further analyze the geographical influence of sprites in the chemistry of the mesosphere. We show in Figure 6 the annual global difference of the mixing ratios of NO$_x$, N$_2$O, HO$_x$ and O$_3$ at 72 km and 50 km altitude between two simulations with and without sprites. In this case, we have assumed that a single sprite injects $10^{20}$ HO$_2$ molecules (Winkler et al., 2021). The maximum increases of the NO$_x$ and N$_2$O mixing ratios at both altitudes are located in the tropical and middle latitudes, where
the area with the largest annual occurrence of sprites. The chemical influence of sprites in the geographical distributions of HO$_x$ and O$_x$ are more complex. The mixing ratios of HO$_x$ and O$_3$ decrease in the areas with a high occurrence rate of sprites at 72 km and 50 km altitude. The HO$_x$ is depleted by the injected molecules of NO$_x$, while O$_3$ is directly depleted by sprites as prescribed by the developed parameterization. The mixing ratio of O$_3$ increases near the poles at 72 km altitude, while its tendency at polar latitudes is not homogeneous at 50 km. At 72 km altitude, the implemented parameterization of sprites
imposes an injection of NO without NO$_2$. The NO injected at 72 km altitude at tropical and middle latitudes is transported polewards and produces O$_3$ in the presence of N. The excess of O$_3$ at 72 km altitude at polar latitudes contributes to a depletion of HO$_x$ molecules. The same relationship between O$_3$ and HO$_x$ can be seen at 50 km altitude, where the inhomogeneous variations of the mixing ratios of HO$_x$ and O$_3$ are spatially correlated.

Figure 7 shows the annual global difference between the mixing ratios of the analyzed chemical species assuming that a
single sprite injects $10^{25}$ HO$_2$ molecules (Yamada et al., 2020) instead of $10^{20}$ HO$_2$ molecules (Winkler et al., 2021). The larger injection of HO$_2$ molecules does not produce any significant difference in the variation of NO$_x$ and N$_2$O between the simulations with and without sprites. However, clear differences of the impact of sprites for the background mixing ratios of HO$_x$ and O$_3$ can be seen (see values and geographical distribution). There is a significant enhancement of the HO$_x$ mixing ratio at 72 km in the regions with the largest occurrence of sprites, due to the direct injection of HO$_2$. The increase of the mixing
ratio of HO$_x$ produces a decrease of the O$_3$ mixing ratio, as O$_3$ molecules are depleted in the conversion between H, OH and HO$_2$ molecules, such as OH + O$_3$ → HO$_2$ + O$_2$ and HO$_2$ + O$_3$ → OH + 2O$_2$ (Jaeglé et al., 2001; Sander et al., 2011). At 50 km altitude, far from the vertical level where the HO$_2$ is injected (above 70 km), the variations of the HO$_x$ and O$_3$ mixing ratios are nearly similar as in the previous case (Figure 6).

Finally, we show in Figure 8 the annual global difference of the mixing ratio of HNO$_3$ and HNO$_4$ at 72 km altitude between
a simulation with sprites (assuming that a single sprite injects $10^{25}$ HO$_2$ molecules (Yamada et al., 2020)) and a simulation without sprites. There is a clear enhancement of the HNO$_4$ mixing ratio in regions with a large occurrence of sprites produced by the reaction between the injected NO$_x$ and HO$_x$ molecules, mainly NO$_2$ + OH → HNO$_3$, NO$_2$ + OH + M → HNO$_3$ + M and NO$_2$ + HO$_2$ → HNO$_4$ (Sander et al., 2011). According to Figure 8, the relative increase of the global HNO$_4$ mixing ratio (0.3%, see Figure 5) is significantly concentrated at tropical latitudes. In particular, sprites potentially represent a non-
negligible source of upper-mesospheric HNO$_4$ at regional scale in South America and Southeastern Asia. In specific regions, the decrease of the HNO$_3$ mixing ratio is likely attributed to the elevated mixing ratio of OH, which interacts with HNO$_3$, leading to its depletion (Sander et al., 2011).




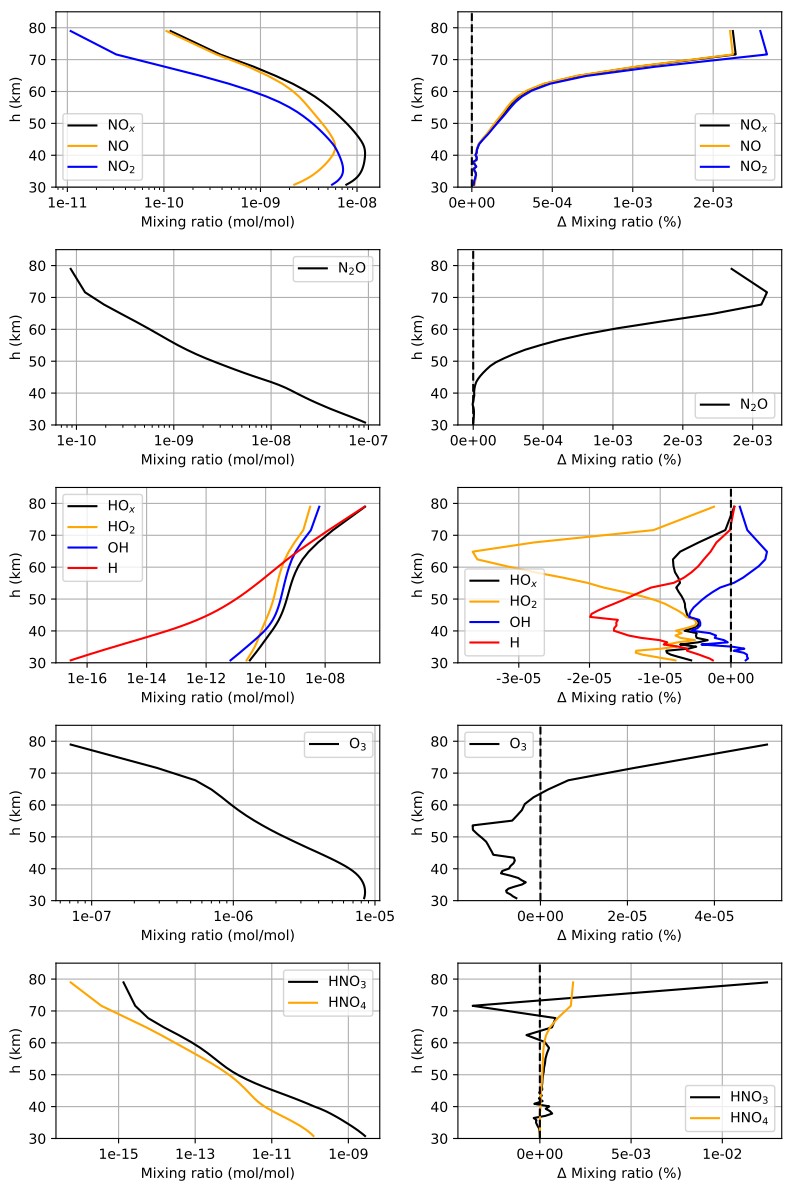

**Figure 4.** First column: Annually (2001) and globally averaged vertical profiles of the mixing ratio of $NO_x$, $N_2O$, $HO_x$, $O_3$, $HNO_3$ and $HO_4$ for a simulation without sprites (CTRL). Second column: Differences (in %) between the annually and globally averaged mixing ratio of the chemical species between the simulation with sprites (SPRI-M) and without sprites (CTRL). In the simulation with sprites, we have assumed that a single sprite can inject $10^{20}$ $HO_2$ molecules (Winkler et al., 2021).



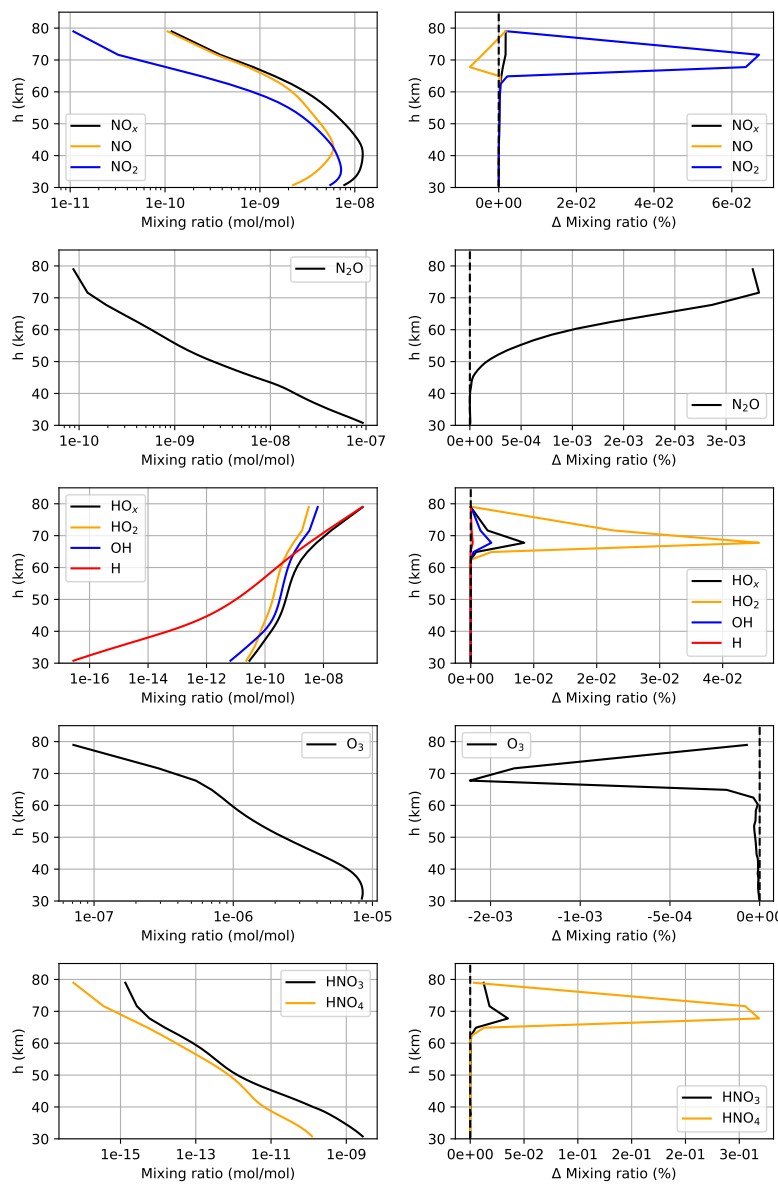

**Figure 5.** Same as Figure 4 but assuming that a single sprite injects $10^{25}$ HO$_2$ molecules (SPRI-SMI simulation) as reported by Yamada et al. (2020).



**Figure 6.** Annually (2001) and globally averaged differences of the $NO_x$, $N_2O$, $HO_x$ and $O_3$ mixing ratios between a simulation with sprites (SPRI-M) and without sprites (CTRL) at 72 km (left) and 50 km (right) altitude. In the simulation with sprites, we have assumed that a single sprite can inject $10^{20}$ $HO_2$ molecules (Winkler et al., 2021).







**Figure 7.** Same as Figure 6 but assuming that a single sprite injects $10^{25}$ HO$_2$ molecules (SPRI-SMI simulation) as reported by Yamada et al. (2020).



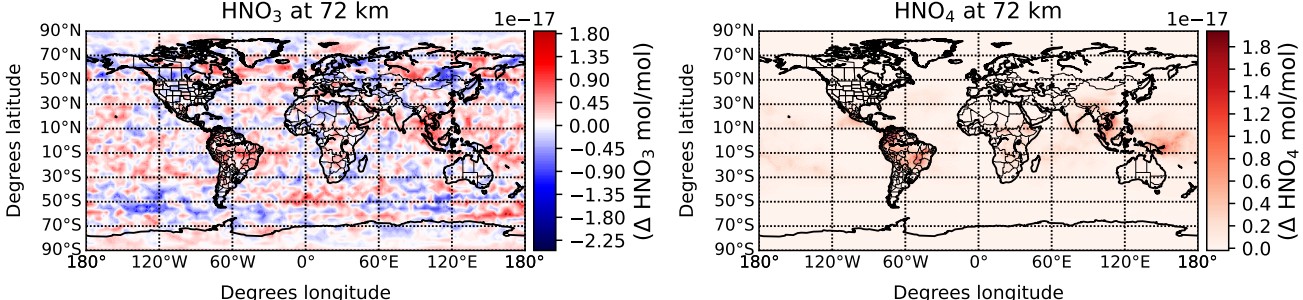

**Figure 8.** Same as Figure 7 but for the reactive nitrogen compounds HNO$_3$ and HNO$_4$.

## 3.3 Regional chemical influence of sprite-streamers: Comparison with NO$_2$ measurements by MIPAS

We now compare the simulated and the observed anomalies of the NO$_2$ mixing ratio at nighttime at the same local hour and
time scale as reported by Arnone et al. (2014). They reported NO$_2$ anomalies over a instantaneous field of view footprint of
30 km × 500 km, an horizontal area that is about 6 times smaller than the area covered by each cell domain of our simulations.
We examine the simulations CTRL and SPRI-M to generate vertical profiles of NO$_2$ and sprite frequency rates for each time
step (720 s) between 22:00 and 23:00 local time during August 2001 within the latitude band 30°S to 20°N. Figure 9(a) shows
the time series of the simulated NO$_2$ mixing ratio time series with and without sprites taking place up to 24 minutes before
300 each given time according to the implemented parameterization of sprites. In total, Figure 9(a) comprises 366,358 data points
for NO$_2$ mixing ratios (black dots), from which 3,408 correspond to post-sprite occurrences (red dots). Following Arnone et al.
(2014), we have removed the NO$_2$ trend in Figure 9(b). The trend is calculated as the average of all the NO$_2$ mixing ratio
data plotted in Figure 9(a). Comparison between the values shown in Figure 9(a,b) and (Arnone et al., 2014, Fig. 2) confirms
that the simulation is producing NO$_2$ mixing ratios at 52 km that are similar to the MIPAS measurements. The anomalies are
305 analyzed in Figure 9(c), showing a positive NO$_2$ mixing ratio anomaly at 52 km altitude, and 24 minutes after the occurrence
of sprites of +0.6 ppbV, which is slightly lower than the +1 ppbV NO$_2$ anomaly reported by Arnone et al. (2014) above
thunderstorms observed by MIPAS. We have performed a t-test on the equality of means of distributions shown in Figure 9(c),
obtaining a $p-value = 1.2 \times 10^{-185}$, which suggests that the difference is statistically significant. In their study, Arnone et al.
(2014) reported NO$_2$ anomalies over an instantaneous field of view with a footprint of 30 km × 1200 km, a size roughly three
310 times smaller than the area covered by each cell domain in our simulations. As a result, it can be expected that the simulated
anomalies are lower than those reported by Arnone et al. (2014).





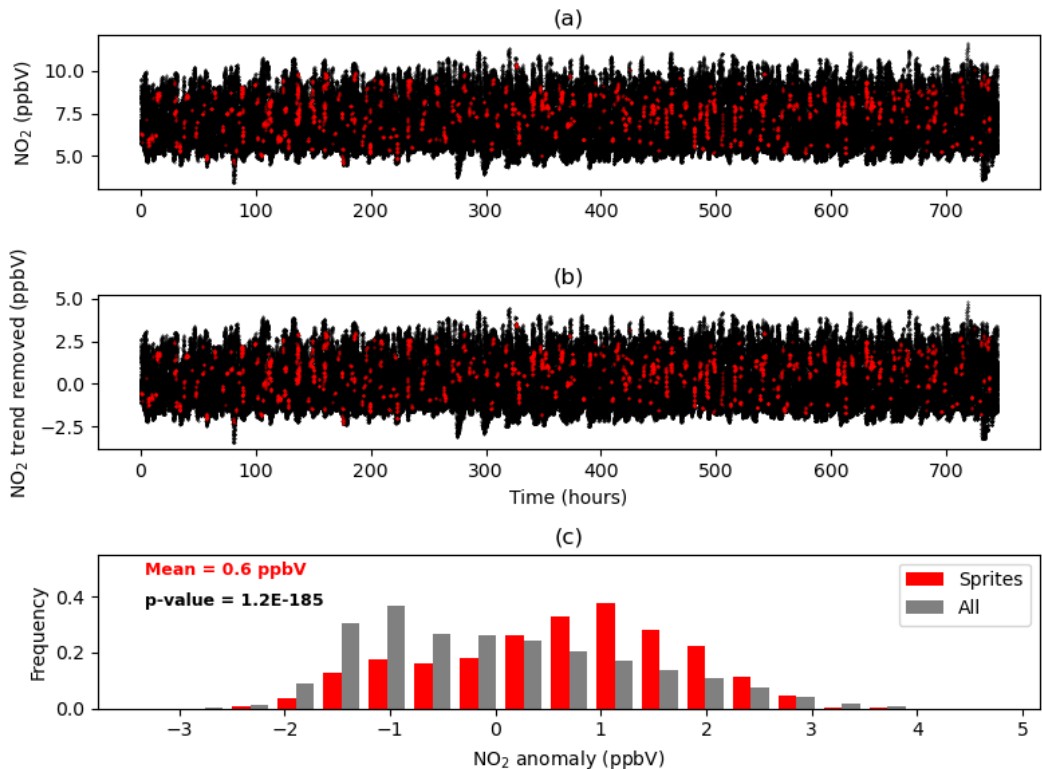

**Figure 9.** (a): Simulated $NO_2$ mixing ratio time series within the latitude band $30°$S to $20°$N between 22:00 and 23:00 local time every 720 s. The horizontal axis represents the time elapsed since August 1, 2001, at 00:00:00 UTC. Black dots denote $NO_2$ mixing ratios when no sprites occurred within the 24-minute window, whereas red dots signify $NO_2$ mixing ratios during the presence of sprites within the same 24-minute period. (b): Same as (a) but after $NO_2$ trend removal. (c): The black distribution shows the simulated $NO_2$ mixing ratio anomalies at 52 km altitude within the latitude band $30°$S to $20°$N between 22:00 and 23:00 local time every 720 s. The red distribution shows the $NO_2$ anomalies for cases in which sprite took place less than 24 minutes before.



## 4 Conclusions

We have developed and implemented in EMAC the first parameterization of sprites based on meteorological variables used as proxy. This parameterization has enabled us to simulate the global annual and seasonal global distributions of sprites and to estimate their sensitivity to climate change. In particular, we have obtained a future increase of the occurrence rate of sprites of 69% in 2091, which is larger than the predicted increase in lightning activity (about 47%). Recent modeling results and space-based measurements have been used to introduce the injection of chemical species by sprites in the model. We have found that the chemical influence of sprites in the mesosphere is not important at a global scale. However, our results indicate that sprites could be a non-negligible (measurable) source of $HNO_4$ at a regional scale, especially in the upper-mesosphere in South America and Southeastern Asia.

The analysis of simulated $NO_2$ mixing ratios above thunderstorms after the occurrence of sprites has confirmed that the anomalies in the nighttime $NO_2$ measurements reported by MIPAS after the occurrence of lightning (Arnone et al., 2014) can be due to sprites. In particular, our simulations indicate an enhancement of +0.6 ppbV of the $NO_2$ mixing ratio above thunderstorms, at 52 km altitude within a 24 minutes window, while the increase reported by Arnone et al. (2014) was +1 ppbV.

The main conclusions of this study are:

1. The developed parameterization of sprites produces a good agreement between the simulated and the observed global distribution of sprites.

2. Implementation of sprites in EMAC produces variations between +0.002% and +0.003% of the mixing ratio of $N_2O$, between +0.002% and +0.06% of $NO_x$, between $-1 \times 10^{-5}$% and 0.0 % of $HO_x$, between -0.002% and $-10^{-5}$% of $O_3$, between +0.01 and +0.05% of $HNO_3$ and between +0.0025% and + 0.3% of $HNO_4$ between 60 km and 80 km altitude in the mesosphere.

3. The globalscale influence of sprites on the chemistry of the atmosphere at a global scale is negligible.

4. Our results confirm that $NO_2$ mixing ratio anomalies reported by MIPAS at 52 km altitude after the occurrence of lightning can be due to sprites.

5. The projected simulation with sprites (RCP6.0) indicates a 54% increase (14% per K) in their occurrence rate at the end of the 21st century, approximately similar to the expected increase of lightning activity.

*Code and data availability.* The Modular Earth Submodel System (MESSy) is continuously developed and applied by a consortium of institutions. The usage of MESSy and access to the source code are licensed to all affiliates of institutions which are members of the MESSy Consortium. Institutions can become a member of the MESSy Consortium by signing the MESSy Memorandum of Understanding. More information can be found on the MESSy Consortium website (http://www.messy-interface.org, last access: 29 September 2023). As the MESSy code is only available under license, the code cannot be made publicly available. The parameterization of sprites has been developed



based on MESSy version 2.54 and will be included in version 2.55. The data of the simulations generated in this study have been deposited in the Zenodo repository Pérez-Invernón et al. (2023)

*Author contributions.* F.J.P.I.: Conceptualization, methodology, validation, formal analysis, investigation, data curation, writing—original
345 draft. F.J.G.V, A.M.R. and P. J.: Conceptualization, methodology, validation, formal analysis, investigation, data curation, writing—review and editing.

*Competing interests.* At least one of the (co-)authors is a member of the editorial board of Atmospheric Chemistry and Physics.

*Acknowledgements.* The project that gave rise to these results received the support of a fellowship from "La Caixa" Foundation (ID 100010434). The fellowship code is LCF/BQ/PI22/11910026 (FJPI). Additionally, this work was supported by grants PID2019-109269RB-
350 C43 (FJPI and FJGV) and PID2022-136348NB-C31 (F.J.P.I. and F.J.G.V) funded by MCIN/AEI/ 10.13039/501100011033 and "ERDF A way of making Europe". AMR acknowledges the financial support from the grant BEVP34A6840 funded by "Ramón Areces Foundation". FJPI and FJGV acknowledge financial support from the grant CEX2021-001131-S funded by MCIN/AEI/ 10.13039/501100011033. PJ acknowledges funding from the Initiative and Networking Fund of the Helmholtz Association through the project "Advanced Earth System Modelling Capacity (ESM)" and from the Helmholtz Association project "Joint Lab Exascale Earth System Modelling (JL-ExaESM)". The
355 content of the paper is the sole responsibility of the author(s) and it does not represent the opinion of the Helmholtz Association, and the Helmholtz Association is not responsible for any use that might be made of the information contained. The high performance computing simulations (HPC) have been carried out on the DRAGO supercomputer of CSIC.



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
