# Peer review of "Gobal and regional chemical influence of sprites: Reconciling modeling results and measurements"

_EGUsphere, 2023_

## Referee Comment (RC1)

The manuscript by Francisco Perez-Invernon and colleagues reports the global chemical impact of streamer discharges in the upper atmosphere, known as sprites, which are caused by particularly intense lightning discharges with continuing current. The results are obtained by using meteorological parameterisations of lightning continuing current in a chemistry-climate model, the subsequent estimation of sprite occurrences and an injection of the associated chemical perturbations based on previous chemical simulations. It is found that some constituents, such as $NO_2$, $HNO_3$, and $HNO_4$ in the middle atmosphere exhibit sizeable concentration changes on regional scales following sprite occurrences and that the number of sprite occurrences would follow a temperature increase caused by climate change. The paper is very well written, logically constructed and easy to follow. The figures and tables are informative and support the text well. However, the authors should consider some clarifications as outlined in the comments and suggestions below.

(1) l18: VLF electromagnetic radiation from lightning discharges is attributed to the return stroke, not to lightning continuing current at ELF/ULF frequencies.

(2) l81: State how many $HO_2$ molecules are 'considerably larger' for comparison.

(3) l86: State which meteorological parameters are used to parametrise sprite activity.

(4) l103: truncation - It might be better to give the high level information first before being more specific, as most readers are unlikely to be familiar with the designation 'T42L90MA'.

(5) l121: 'sprite density' seems more appropriate, as they are streamers rather than flashes.

(6) l127: The relevant parameter is the charge moment change and the time over which the charge moment is reached, rather than solely the duration of the continuing current (Cummer, GRL, 2001, Fig. 3)

(7) l128: Explain how the lightning continuing current density is computed.

(8) l129: 'Nighttime' and the 'absence of solar radiation' is a rather gradual process ranging from civil to nautical, and astronomical twilight.

(9) l136: Sprites are generated by the quasi-static removal of a relatively large lateral charge distribution where lightning continuing current is perhaps the most prominent indicator, but not necessarily the only mechanism that contributes, eg consider IC lightning.

(10) l151: State the physical quantities that are interpolated and extrapolated.

(11) l164: ELF radio measurements are more indicative ~1000 streamers (Qin, GRL, 2012, L22803, p4).

(12) l230: Explain in more detail what is meant by 'good agreement', which is not obvious.

(13) l236: It is not clear why an annual global average is a meaningful measure, given the high level of detail provided by the simulations. Lightning mainly follows solar insolation over the continents, whereas ~71% of the earth is covered by water and the summer season covers ~25% of the year.

(14) l261, Fig. 6 caption: Explain how the differences are calculated, why % is not used and whether the scaling $\sim10^{-15}$ is possibly numerical noise. What is the physical reason for the banded structures shown for $NO_x$ and $N_2O$?

(15) l286, Fig. 8: The Congo basin in Africa tends to be the largest contributor to global lightning activity such that is somewhat unclear why South America and Southeast Asia appear to be more prominent on these global maps. Is there perhaps an association with aerosols?

(16) l298: Explain why an asymmetric latitude range is chosen.

(17) l308: It is not clear what the meaning of the t-test is, given that the data shown in Fig. 9 is obviously not normal distributed. In this case, a significance level should be stated, if it can be evaluated at all.

(18) Fig 9, caption: Reword to 'gray distribution'.

l328: Specify where these numbers given in % have been reported in the text or the figures.

l332: Delete the first 'globalscale'.

---

## Author Comment (AC1)

We appreciate the reviewer for the time spent revising this manuscript and for providing valuable comments. We have addressed the reviewer's comments below (in blue font).

In addition, we want to point out that we have re-run the chemical simulations because we detected a small error in the code implementation related to the vertical profile of $NO_2$ and $N_2O$ injection. Despite this change, the results obtained are very similar.

The manuscript by Francisco Perez-Invernon and colleagues reports the global chemical impact of streamer discharges in the upper atmosphere, known as sprites, which are caused by particularly intense lightning discharges with continuing current. The results are obtained by using meteorological parameterisations of lightning continuing current in a chemistry-climate model, the subsequent estimation of sprite occurrences and an injection of the associated chemical perturbations based on previous chemical simulations. It is found that some constituents, such as NO2, HNO3, and HNO4 in the middle atmosphere exhibit sizeable concentration changes on regional scales following sprite occurrences and that the number of sprite occurrences would follow a temperature increase caused by climate change. The paper is very well written, logically constructed and easy to follow. The figures and tables are informative and support the text well. However, the authors should consider some clarifications as outlined in the comments and suggestions below.

We thank the reviewer for his/her encouraging comments.

(1)l18: VLF electromagnetic radiation from lightning discharges is attributed to the return stroke, not to lightning continuing current at ELF/ULF frequencies.

Done:

"The quasi-electrostatic field primarily generates electromagnetic radiation within the Extremely Low Frequency (ELF) **and the Ultra low frequency (ULF)** electromagnetic spectra, which result from continuing currents during discharges lasting several tens or hundreds of milliseconds ..."

(2)l81: State how many HO2 molecules are 'considerably larger' for comparison.

Done:

"... which is considerably larger than the production of **$10^{20}$ molecules of HO$_2$** estimated by Winkler et al. (2021)."

(3)l86: State which meteorological parameters are used to parametrise sprite activity.

Done:

"In this study, we present the first parameterization of sprites based on the proxy meteorological parameter vertical velocity at the 450 hPa level for sprite activity"

(4)l103: truncation - It might be better to give the high level information first before being more specific, as most readers are unlikely to be familiar with the designation 'T42L90MA'.

Done:

"The model is operated with a triangular truncation of the spectral resolution at wave number 42, corresponding to a quadratic Gaussian grid with a resolution of 2.8° in both latitude and longitude. It comprises 90 vertical levels, extending up to the 0.01 hPa pressure level, and employs a time step length of 720 seconds, as described by Jöckel et al. (2016) for the T42L90MA resolution."

(5)l121: 'sprite density' seems more appropriate, as they are streamers rather than flashes.

Done. We changed "sprite flash density" by "sprite density" everywhere in the manuscript.

(6)l127: The relevant parameter is the charge moment change and the time over which the charge moment is reached, rather than solely the duration of the continuing current (Cummer, GRL, 2001, Fig. 3)

We agree with the reviewer and we have changed the text accordingly:

"**Sprites are generated by the charge moment change resulting from lightning, along with the duration over which the charge moment is attained** (Cummer, 2001). Lightning flashes with continuing currents, such as LCC(>18 ms) lightning, can induce a substantial charge moment change and a high quasi-electrostatic field in the mesosphere (Gamerota et al., 2011). Consequently, this process can trigger the initiation of sprites"

(7)l128: Explain how the lightning continuing current density is computed.

Done:

"Therefore, we use the LCC(>18 ms) lightning density, computed by the LNOX submodel of MESSy based on the vertical velocity at the 450 hPa pressure level, as a proxy for the occurrence of sprites."

(8)l129: 'Nighttime' and the 'absence of solar radiation' is a rather gradual process ranging from civil to nautical, and astronomical twilight.

We define "nighttime" as the time when the Sun is below the horizon. We added:

"In addition, we imposed that nighttime sprites can only be produced after sunset, **when the Sun is below the horizon**"

(9)l136: Sprites are generated by the quasi-static removal of a relatively large lateral charge distribution where lightning continuing current is perhaps the most prominent indicator, but not necessarily the only mechanism that contributes, eg consider IC lightning.

We have added this to the manuscript. We think this reasoning connects very well with the references to Greenberg et al. (2009) and Lu et al. (2017):

"Sprites are generated by the quasi-static removal of a relatively large lateral charge distribution where lightning continuing current is perhaps the most prominent indicator, but not necessarily the only mechanism that contributes."

(10)l151: State the physical quantities that are interpolated and extrapolated.

We have stated how we interpolate/extrapolate the physical quantities NO, $NO_2$, $N_2O$ and $O_3$ per meter. More details can be found in Malagón-Romero et al. (2023):

"In particular, we estimate the production per meter of NO, $NO_2$, and $N_2O$ molecules, as well as the depletion per meter of $O_3$ molecules, in the altitude range from 49.75 km to 50 km and from 67 km to 75 km. Subsequently, we extrapolate the production or removal of molecules from 45 km to 49.75 km and interpolate the production or removal of molecules from 50 km to 75 km."

(11)l164: ELF radio measurements are more indicative ~1000 streamers (Qin, GRL, 2012, L22803, p4).

While ELF radio measurements suggest the presence of over 1000 streamers per individual sprite (Qin et al., 2012), it is essential to recognize that the characteristics and production of chemical species within streamers can be heterogeneous (Stenbaek-Nielsen et al., 2013). Consequently, multiplying the injection of chemical species per streamer by the total number of streamers may lead to inaccuracies. To address this, we conduct a comparison between the simulated and observed total number of photons, allowing us to estimate the production of chemical species by observed sprites based on simulation results.

We have added this to the manuscript.

(12)l230: Explain in more detail what is meant by 'good agreement', which is not obvious.

We refer to the observed and the simulated spatial maximum of sprite density:

"There is a good agreement between the simulated (Figure 3) and the reported (Sato and Fukunishi, 2003, Fig. 4(a)) spatial maximum of sprite density in JJA near the Equator and in Southeastern Asia"

(13)l236: It is not clear why an annual global average is a meaningful measure, given the high level of detail provided by the simulations. Lightning mainly follows solar insolation over the continents, whereas ~71% of the earth is covered by water and the summer season covers ~25% of the year.

Global average is chosen to determine the role of a phenomenon in the chemical composition of the atmosphere, as in the case of lightning (e. g., Schumann and Huntrieser (2007), Gordillo-Vázquez et al. (2019)). We choose global average to show the role of sprites in the chemistry composition of the atmosphere without any seasonal/regional effect. In Section 3.3, we investigate the regional effect during August – December in order to compare with measurements.

(14)l261, Fig. 6 caption: Explain how the differences are calculated, why % is not used and whether the scaling ~10-15 is possibly numerical noise. What is the physical reason for the banded structures shown for NOx and N2O?

The differences are calculated as the mixing ratio in the simulation with sprites minus the mixing ratio in the simulation without sprites, as it is already explained in the caption.

Please note that these figures complement to Figures 4 and 5, given in % and showing the absolute mixing ratio at 72 km altitude. Combining Figures 4-8 gives the complete view of the role of sprites in the chemical composition of the atmosphere. In addition, please note that we have applied the QCTM mode, which removes the numerical noise.

The banded structures of Figure 6 are due to the location of sprites. Fig. 1 shows that sprites take place at the regions where we found the increases of $NO_x$ and $N_2O$ mixing ratios. This is already stated in the manuscript:

"The maximum increases in $NO_x$ and $N_2O$ mixing ratios at both altitudes are observed in the tropical and middle latitudes. This region coincides with the area that experiences the largest annual occurrence of sprites."

(15)l286, Fig. 8: The Congo basin in Africa tends to be the largest contributor to global lightning activity such that is somewhat unclear why South America and Southeast Asia appear to be more prominent on these global maps. Is there perhaps an association with aerosols?

Although the Congo basin in Africa tends to be the largest contributor to global lightning, it is not the case for sprites (see Fig. 1). The reason is that Africa is not the largest contributor to lightning with CC (see for example Bitzer 2017, Pérez-Invernón et al., 2022).

(16)l298: Explain why an asymmetric latitude range is chosen.

This latitude range coincides with that studied by Arnone et al. (2008). We selected this specific range for comparison with our observations. We erroneously cited Arnone et al. (2014) instead of Arnone et al. (2008) in this paragraph.

(17)l308: It is not clear what the meaning of the t-test is, given that the data shown in Fig. 9 is obviously not normal distributed. In this case, a significance level should be stated, if it can be evaluated at all.

As the distributions are not normal, we have changed the method to determine the risk of coincidence of both distributions:

"We computed the 95% confidence intervals for the mean and median of the distributions using a bootstrap method (Efron and Tibshirani, 1994) with 5000 resamples. The calculated confidence intervals are (-0.0022 ppbV, 0.0023 ppbV) and (0.5334 ppbV, 0.5843 ppbV) for the means of the distribution of simulated $NO_2$ mixing ratio anomalies at 52 km altitude and the distribution of $NO_2$ anomalies for cases in which sprites occurred less than 24 minutes before, respectively. Notably, these intervals do not overlap, suggesting significant differences between the distributions."

Please note that we have now expanded the period to August – December 2001, as in Arnone et al. (2008).

(18)Fig 9, caption: Reword to 'gray distribution'.

Done.

l328: Specify where these numbers given in % have been reported in the text or the figures.

Done (in Section 3.2).

l332: Delete the first 'globalscale'.

Done.

---

## Author Comment (AC2)

We appreciate the reviewer for the time spent revising this manuscript and for providing valuable comments. We have addressed the reviewer's comments below (in blue font).

In addition, we want to point out that we have re-run the chemical simulations because we detected a small error in the code implementation related to the vertical profile of $NO_2$ and $N_2O$ injection. Despite this change, the results obtained are very similar.

**Reviewer 2**
The paper presents new and interesting results on the global chemical effects of sprites. It is well written and focused. It addresses relevant scientific questions within the scope of ACP, and deserves to be published. For the final revision, I'd like to ask the authors to take into consideration these points:

We thank the reviewer for his/her encouraging comments.

Line 132: Please comment on the ratio of LCC-flashes to all flashes. How does your 20% relate to the 1/1000 sprite-to-flash estimate by Arnone et al. (2014)?

We (Pérez-Invernón et al., 2022) reported that among $3.5 \times 10^6$ flashes recorded by ISS-LIS over one year, only $2.6 \times 10^4$ flashes have a continuing current lasting more than 18 ms (about 7.4/1000 flashes). We set that only 20% can produce sprites (1.5/1000). Finally, we set that only flashes taking place during nighttime can produce sprites. Therefore, our 20% relates quite well with the 1/1000 sprite-to-flash estimate by Arnone et al. (2014).

Added:

"Arnone et al. (2014) estimated that about 1/1,000 flashes could produce a sprite, while Pérez-Invernón et al. (2021) found that 7.4/1,000 flashes reported by LIS during one year have a continuing current lasting more than 18 ms. Therefore, the approximation of 1 sprites per 20% nighttime LCC(>18 ms) lightning flash is of the same order as the 1/1,000 sprite-to-flash estimate by Arnone et al. (2014)."

Table 1 states 'SPRI-M … HOx by Malagón-Romero et al (2023)' Are there HOx production estimates in that paper? If so, please give numbers like for the cases Yamada et al. / Winkler et al.. If not, please correct the reference.

No, there were not $HO_x$ production estimates in that paper. We have changed the reference to Winkler et al. (2021).

Line 144: As far as I see, that wasn't an 'electrodynamic' model?

We agree, because this model is 0D. We have removed "electrodynamic".

Line 154: After 'single sprite' add 'streamer'.

Done.

Section 3.2: Please compare your results to the model results of Arnone et al. (2014).

Added:

"We obtained a marginal increase of approximately 0.007% in the background concentration of $NO_x$ at an altitude of 70 km. This increment is notably lower than the perturbation estimated by Arnone et al. (2014) due to sprites, which falls within the range of 2% to 20%. The variance in results can be attributed to the disparity in assumptions made by Arnone et al. (2014), who considered an injection of $NO_x$ molecules ranging from $1.5 \times 10^{23}$ to $1.5 \times 10^{24}$. In contrast, our study assumes a more conservative injection of $6.46 \times 10^{22}$ $NO_x$ molecules. In addition, the sprite-$NO_x$ perturbation profile in this study is linear between the altitudes 45 km and 80 km, while the profile adopted by Arnone et al. (2014) peaks at about 65 km altitude."

Figure 6 caption: 'globally averaged' might be wrong.

Done.